# Humoral and cellular immune responses to COVID-19 mRNA vaccines in immunosuppressed liver transplant recipients
Takuto Nogimori [1,8], Yuta Nagatsuka[1,2,8], Shogo Kobayashi[2] ✉, Hirotomo Murakami[1,2], Yuji Masuta[1], Koichiro Suzuki[3], Yoshito Tomimaru[2], Takehiro Noda[2], Hirofumi Akita[1,4,5], Shokichi Takahama [1], Yasuo Yoshioka[3], Yuichiro Doki[2], Hidetoshi Eguchi[2] & Takuya Yamamoto [1,5,6,7] ✉

## Abstract

**Background** Liver transplant recipients (LTRs) are at a high risk of severe COVID-19 owing to immunosuppression and comorbidities. LTRs are less responsive to mRNA vaccines than healthy donors (HDs) or other immunosuppressed patients. However, the disruption mechanism in humoral and cellular immune memory responses is unclear.

**Methods** We longitudinally collected peripheral blood mononuclear cells and plasma samples from HDs ($n = 44$) and LTRs ($n = 54$) who received BNT162b2 or mRNA-1273 vaccines. We measured the levels of anti-receptor-binding domain (RBD) antibodies and spike-specific CD4$^+$ and CD8$^+$ T-cell responses.

**Results** Here, we show that the induction of anti-RBD IgG was weaker in LTRs than in HDs. The use of multiple immunosuppressive drugs is associated with lower antibody titers than only calcineurin inhibitor, and limits the induction of CD4$^+$ T-cell responses. However, spike-specific CD4$^+$ T-cell and antibody responses improved with a third vaccination. Furthermore, mRNA vaccine-induced spike-specific CD8$^+$ T cells are quantitatively, but not qualitatively, limited to LTRs. Both CD4$^+$ and CD8$^+$ T cells react to omicron sublineages, regardless of the presence in HDs or LTRs. However, there is no boosting effect of spike-specific memory CD8$^+$ T-cell responses after a third vaccination in HDs or LTRs.

**Conclusions** The third mRNA vaccination improves both humoral responses and spike-specific CD4$^+$ T-cell responses in LTRs but provides no booster effect for spike-specific memory CD8$^+$ T-cell responses. A third mRNA vaccination could be helpful in LTRs to prevent severe COVID-19, although further investigation is required to elicit CD8$^+$ T-cell responses in LTRs and HDs.

## Plain language summary

People with a liver transplant don't have as strong an immune response to COVID-19 vaccines as healthy people. This study investigates how these individuals produce protective proteins, called antibodies, and CD4 and CD8 T cell immune responses. CD4 T cells are responsible for commanding the immune response and CD8 T cells for remembering and fighting the virus in future. We found that liver transplant recipients have a weaker ability to produce antibodies after vaccination, which is even more noticeable in those taking drugs to prevent transplant rejection. While a third vaccine dose improves their ability to produce antibodies, and to have a CD4 T cell response, it doesn't boost the CD8 T cell response. In summary, an extra vaccine dose can strengthen the immune response in liver transplant recipients but doesn't improve some aspects of their immune memory.

[1]Laboratory of Precision Immunology, Center for Intractable Diseases and ImmunoGenomics, National Institutes of Biomedical Innovation, Health and Nutrition, Osaka 567-0085, Japan. [2]Department of Gastroenterological Surgery, Graduate School of Medicine, Osaka University, Osaka 565-0871, Japan. [3]The Research Foundation for Microbial Diseases of Osaka University (BIKEN), Osaka 565-0871, Japan. [4]Department of Gastroenterological Surgery, Osaka International Cancer Institute, Osaka 540-0008, Japan. [5]Laboratory of Translational Cancer Immunology and Biology, Next-generation Precision Medicine Research Center, Osaka International Cancer Institute, Osaka 540-0008, Japan. [6]Department of Virology and Immunology, Graduate School of Medicine, Osaka University, Osaka 565-0871, Japan. [7]Laboratory of Aging and Immune Regulation, Graduate School of Pharmaceutical Sciences, Osaka University, Osaka 565-0871, Japan. [8]These authors contributed equally: Takuto Nogimori, Yuta Nagatsuka. ✉e-mail: skobayashi@gesurg.med.osaka-u.ac.jp; yamamotot2@nibiohn.go.jp

Severe acute respiratory syndrome coronavirus 2 (SARS-CoV-2) emerged in late 2019 and caused a respiratory disorder known as coronavirus disease 2019 (COVID-19)[1]. Although most infected patients experienced mild symptoms, older adults and immunosuppressed patients, such as transplant recipients and autoimmune disease patients, are at risk of developing severe COVID-19[2,3]. An international database study revealed that the rate of ICU admission and invasive ventilation required after SARS-CoV-2 infection was significantly higher in liver transplant recipients (LTRs) than in healthy individuals[4]. Age, serum creatinine level, and non-liver cancer status are associated with post-infection mortality in LTRs, indicating that LTRs require adequate infection prevention methods[4].

As a preventive method against COVID-19, mRNA vaccines are believed to be the most effective. Currently, two or more doses (three or four) of mRNA vaccinations are being administered worldwide[5]. Two doses of mRNA vaccines reduce the possibility of SARS-CoV-2 infection and COVID-19-related deaths by 64% and 87%, respectively, in LTRs[6]. However, the use of multiple immunosuppressive drugs, especially mycophenolate mofetil or steroids, is a risk factor for reduced antibodies in LTRs after two mRNA vaccine doses[7,8]. Although two doses of mRNA vaccines effectively prevent SARS-CoV-2 infection, their effects were weaker in LTRs than in healthy individuals or patients with other immunosuppressive conditions[9]. Additionally, three mRNA vaccine doses can improve antibody induction in LTRs[10], although they remain inadequate for LTRs to induce a strong neutralizing activity.

Clinically, immunosuppressive drugs used in LTRs include calcineurin inhibitors (CNIs), such as tacrolimus and cyclosporine, mycophenolate mofetil, everolimus, and steroids[11]. CNIs suppress immune responses by inhibiting initial T-cell activation[12,13]. Therefore, the unresponsiveness of LTRs to antibody induction after mRNA vaccination could be caused by the suppression of T-cell responses. However, studies on mRNA vaccine-induced changes in T-cell responses in LTRs are limited. Moreover, many studies have been limited to verification at specific time points, such as after the second or third vaccination, and insights into the changes in antibody titers and memory T-cell responses over time are insufficient.

A bivalent mRNA vaccine has been designed against an Omicron strain[14] and induces the production of antibodies against BA.5 more efficiently than the monovalent mRNA vaccine against the Wuhan-1 strain[15–17]. However, the neutralizing activity induced by mRNA vaccines is limited in newly emerged strains, such as BA.2.75.2, BQ.1.1, XBB, and XBB.1, as SARS-CoV-2 can evade neutralizing antibodies via mutations in the spike protein[18]. In other words, inducing T-cell responses that can react to mutant strains, rather than relying on neutralizing antibodies, is key to preventing future infections due to emerging mutant strains. Nevertheless, whether mRNA vaccine-induced T-cell responses in immunosuppressed LTRs are reactive to Omicron sublineages is unclear, although mRNA vaccine-induced antigen-specific T-cell responses in healthy individuals could be cross-reactive against the SARS-CoV-2 Omicron strain (BA.1)[19,20].

In this study, we longitudinally collect peripheral blood mononuclear cells (PBMCs) and plasma samples from healthy donors (HDs) and LTRs who received BNT162b2 or mRNA-1273 vaccines. We evaluate the mRNA vaccine-induced humoral and cellular immune memory responses over time and demonstrate that the generation of anti-RBD IgG antibodies is less effective in LTRs compared to HDs. The employment of a combination of immunosuppressive medications results in reduced antibody levels as opposed to using solely calcineurin inhibitors, and this adversely impacts the development of CD4$^+$ T-cell responses. Nevertheless, the response of spike-specific CD4$^+$ T cells and antibodies is enhanced following a third dose of the mRNA vaccine. Additionally, while the quantity of mRNA vaccine-elicited spike-specific CD8$^+$ T cells in LTRs is lower, their functionality remains unchanged. Both CD4$^+$ and CD8$^+$ T cells show reactivity to omicron subvariants, irrespective of their occurrence in HDs or LTRs. However, no enhancement in the memory responses of spike-specific CD8$^+$ T cells is observed after a third vaccine dose in either HDs or LTRs.

## Methods

### Study participants
A total of 98 individuals (44 healthy donors and 54 LTRs) were recruited from Osaka University, Japan. Blood samples were collected, and peripheral blood mononuclear cells (PBMCs) were isolated via density gradient centrifugation using a BD Vacutainer cell preparation tube (CPT) containing sodium heparin (Becton, Dickinson and Co., Franklin Lakes, NJ, USA) according to the manufacturer's instructions. PBMCs were immersed in fetal bovine serum containing 10% dimethyl sulfoxide and stored in liquid nitrogen until analysis. The donor information used in this study is presented in Table 1.

### SARS-CoV-2 spike-specific antibody detection
Plasma levels of total IgG-targeting SARS-CoV-2 spike receptor-binding domain (RBD)-specific antibodies were determined by ELISA[20]. Recombinant spike RBD proteins (Wuhan-1, BA.1, BA.2, BA.5, BQ.1.1, XBB, BA.4.6, and BA.2.75) were obtained from SinoBiological (Beijing, China). To calculate RBD-specific antibody titers, 96-well plates were coated with RBD protein and incubated overnight at 4 °C. The plates were then washed and incubated for 1 h with blocking buffer, then washed again, and incubated with diluted plasma samples for 2 h at 25 °C. The plates were washed and incubated with biotinylated anti-human total IgG (BD Biosciences, San Jose, CA, USA) for 1 h. The plates were washed and incubated with horseradish peroxidase-conjugated streptavidin (Thermo Fisher Scientific, Waltham, MA, USA) for 1 h at 25 °C. The plates were washed and incubated with the TMB peroxidase substrate (KPL, Gaithersburg, MD, USA) for color development. After 10 min, 2 mol/L H$_2$SO$_4$ was added to each well to stop the reaction. Antibody expression was measured by determining the optical density at 450 nm using an Epoch 2 Microplate Spectrophotometer (Agilent, Santa Clara, CA, USA). The antibody endpoint titer was determined using a cut-off value of 0.3. The cutoff value of OD = 0.3 was determined based on the OD values of plasma from unvaccinated individuals used as a negative control, specifically by adding twice the standard deviation to the average OD value.

### Neutralization assay of pseudotyped virus
To determine the pseudotyped virus neutralization titer (pVNT) of the vaccinated donors' plasma, HEK-293A cells expressing ACE2 and TMPRSS2 were seeded in 96-well plates. After 24 h, the human plasma was diluted 2- or 4-fold, starting at 1:2, and incubated with SARS-CoV-2 pseudotyped virus at 37 °C for 1 h. After incubation, a mixture of plasma and the pseudotyped virus was added to each well. After 24 h, luciferase activity was measured using EnSpire (PerkinElmer, Waltham, MA, USA). pVNT50 was defined as the plasma dilution that achieved 50% inhibition of pseudotyped virus infection using Prism software (GraphPad Software, Boston, MA, USA).

### Flow cytometry analysis
To analyze SARS-CoV-2 spike-specific T cells, we performed surface and intracellular cytokine staining of CD4$^+$ and CD8$^+$ T cells[20,21]. Briefly, PBMCs were incubated in 1 mL RPMI 1640 medium containing 50 U/mL benzonase nuclease (Millipore, Darmstadt, Germany), 10% fetal bovine serum, and penicillin-streptomycin for 2 h. Next, cells were incubated in 200 μL of medium with or without peptides (17-mers overlapping by 11 residues) corresponding to the full-length SARS-CoV-2 spike, at a final concentration of 2 μg/mL of each peptide, for 30 min. Thereafter, 0.2 μL BD GolgiPlug and 0.14 μL BD GolgiStop (both from BD Biosciences) were added and incubated for 5.5 h. The cells were then stained using a LIVE/DEAD Fixable Blue Dead Cell Stain Kit (Thermo Fisher Scientific) and anti-CD3 (SP34-2, 1:100 dilution), anti-CD8 (RPA-T8, 1:400 dilution), anti-CD4 (L200, 1:100 dilution), anti-CD45RO (UCHL1, 1:200 dilution), anti-CD27 (O323, 1:100 dilution), and anti-CD57 (NK-1, 1:2000 dilution) antibodies. After fixation and permeabilization using a Cytofix/Cytoperm kit (BD Biosciences), the cells were stained with anti-CD154 (TRAP1, 1:14 dilution), anti-4-1BB

**Article**

**Table 1 | Donor characteristics enrolled in this study**

| Caracteristics | Liver transplant recipients ($n = 54$) | Healthy donors ($n = 44$) |
|---|---|---|
| **Demographic** | | |
| Age, median years (interquartile range) | 65 (56.3–70.8) | 36 (34–44.3) |
| Male (%) | 28 (51.9) | 35 (79.5) |
| Female (%) | 26 (48.1) | 9 (20.5) |
| **Vaccine** | | |
| **2nd** | | |
| Pfizer/BNT162b2 | 42 (77.8) | 25 (56.8) |
| Moderna/mRNA-1273 | 12 (22.2) | 19 (43.1) |
| **3rd** | | |
| Pfizer/BNT162b2 | 29 (53.7) | 25 (56.8) |
| Moderna/mRNA-1273 | 25 (46.3) | 19 (43.1) |
| **Laboratory values (Median, Interquartile range)** | | |
| WBC ($\times 10^9$/L) | 5.2 (4.5–6.8) | NA |
| Lymphocytes ($\times 10^9$/L) | 1.5 (1.0–1.9) | NA |
| Neutrophils ($\times 10^9$/L) | 3.2 (2.7–4.0) | NA |
| Monocytes ($\times 10^9$/L) | 0.32 (0.27–0.45) | NA |
| CRP (mg/dL) | 0.08 (0.043–0.13) | NA |
| eGFR (mL/min) | 51.6 (41.6–65.0) | NA |
| HbA1c (%) | 5.9 (5.3–6.6) | NA |
| **Immunotherapy ($n$, %)** | | |
| Monotherapy | 23 (42.6) | NA |
| Tacrolimus | 19 (35.2) | NA |
| Cyclosporine | 4 (7.4) | NA |
| CNI + MMF | 12 (22.2) | NA |
| CNI + mTORi | 2 (3.7) | NA |
| CNI + steroid | 11 (20.4) | NA |
| CNI + MMF + steroid | 5 (9.3) | NA |
| CNI + mTORi + steroid | 1 (1.9) | NA |
| Entecavir | 7 (13.0) | NA |

(4B4-1, 1:400), anti-CD69 (FN50, 1:67), anti-IFN-γ (4 S. B3, 1:100), anti-TNF (MAb11, 1:50 dilution), anti-IL-2 (MQ-17H12, 1:400 dilution), anti-IL-4 (8D4-8, 1:40 dilution), anti-IL-13 (JES10-5A2, 1:40 dilution), anti-granzyme A (CB9, 1:400 dilution), anti-granzyme B (GB11, 1:2000), and anti-perforin (B-D45, 1:67 dilution) antibodies. The cells were then analyzed using a BD FACSymphony A5 flow cytometer (BD Biosciences) and FlowJo v. 10.8.1. After gating live single T cells based on the forward scatter area and height (FSC-A and -H), side scatter area (SSC-A), live/dead cell exclusion, and CD3 staining, PBMCs were separated into CD4$^+$ and CD8$^+$ T cells. Subsequently, CD4$^+$ and CD8$^+$ T cells were further divided into memory phenotypes based on their CD27 and CD45RO expression. For spike-specific CD4$^+$ and CD8$^+$ T cells, memory cells were gated based on the expression of CD154 and 4-1BB and CD69 expression, respectively. We defined CD154$^+$CD4$^+$ T cells expressing IFN-γ, TNF, or IL-2 as Th1 cells and those expressing IL4 or IL-13 as Th2 cells. Frequencies of CD154$^+$CD4 T cells, Th1, Th2 and CD69$^+$4-1BB$^+$CD8 T cells were calculated by subtracting background of unstimulated samples (DMSO). Positive responses were defined if there was a reactivity of 0.01% or more after background subtraction from the unstimulated condition.

## Statistics
Indivisual endpoint titers, pVNT$_{50}$ and FACS data are shown as median with interquartile range. Statistical analyses were performed with GraphPad Prism 9, Spice 6.1, and R programming language. Mann–Whitney $U$ test

and Wilcoxon matched-pairs signed-rank test were used for comparisons of groups. Correlations were calculated using a nonparametric Spearman's rank test. Multivariable logistic regression model was used for prediction of the relationships between dependent and independent variables. In all figures, $P$ values are indicated by *$P < 0.05$; **$P < 0.01$; ***$P < 0.001$; ****$P < 0.0001$.

## Study approval
The study protocol and procedures were reviewed and approved by the Institutional Ethics Committees of the National Institutes of Biomedical Innovation, Health, and Nutrition (approval nos. 137, 505, and 117-4), Osaka, Japan, and Osaka University (approval no. 21195), Osaka, Japan, and complied with the 1975 Declaration of Helsinki. All the participants provided written informed consent to participate in the study.

## Reporting summary
Further information on research design is available in the Nature Portfolio Reporting Summary linked to this article.

## Results
### Anti-RBD IgG titers and plasma neutralizing activity induced by COVID-19 mRNA vaccination in HDs and LTRs
We enrolled 44 HDs and 54 LTRs to comprehensively evaluate mRNA vaccine-induced antibodies and cellular immune responses (Table 1). The mRNA vaccines, Pfizer BNT162b2 or Moderna mRNA-1273 were investigated. Blood samples were obtained at five-time points: before vaccination, 1, 3, and 6 months after the second vaccination, and 1 month after the third vaccination (Fig. 1a).

All LTRs were administered CNIs, such as tacrolimus or cyclosporine. Some LTRs took additional medications, such as the metabolic antagonist MMF, a steroid, or the mTOR inhibitor everolimus. Specifically, 23, 12, 2, 11, 5, and 1 LTRs had taken only a CNI; CNI and MMF; CNI and everolimus; CNI and a steroid; CNI, MMF, and a steroid; and CNI, everolimus, and a steroid, respectively. Seven LTRs received entecavir, a drug used to treat hepatitis B, and immunosuppressive therapy.

Anti-RBD antibody titers in LTRs were significantly lower than those in HDs at all time points (Fig. 1b) ($p < 0.0001$ at 1 and 3 months, $p = 0.0005$ at 6 months after the second vaccination, $p = 0.0002$ after the third vaccination). Anti-RBD antibody titers in all HDs exceeded the WHO standard (dashed line, 1000 U/mL); however, 53.2% of LTRs had anti-RBD antibody titers below the WHO standard at 1 month after the second vaccination. However, anti-RBD antibody titers in 92.2% of the LTRs after the third vaccination exceeded the WHO standard, suggesting that effective immune responses can be achieved in immunosuppressed LTRs by the third vaccination.

Interestingly, the variability in antibody levels among LTRs was wide compared with that in HDs. Therefore, we aimed to identify the factors that affect the variability in antibody production in LTRs. LTRs that obtained anti-RBD antibody levels higher and lower than the median value of antibody titers in HDs after the third vaccination were categorized as strong and weak responders, respectively. We conducted a multiple logistic regression analysis with clinical parameters (Fig. 1c), suggesting that taking multiple drugs decreased antibody levels ($p = 0.0048$, OR = 0.0285).

We regrouped LTRs for comparison between LTRs taking only a CNI and taking a CNI and more drugs (CNI+other drug(s)) (Fig. 1d). There was no difference in the antibody titers between the CNI group and HDs after the third vaccination. Contrarily, antibody titers were significantly lower in the CNI+other drug(s) group than in the HDs and the CNI group ($p < 0.0001$ among HDs vs. CNI+other drug(s), $p < 0.0001$ among CNI vs. CNI+other drug(s)). However, the anti-RBD antibody titers after the third vaccination in the CNI+other drug(s) group were the same as those in HDs 1 month after the second vaccination (Fig. 1e; $p = 0.3255$). After the second vaccination, anti-RBD antibodies in plasma were induced in 49 of 54 LTRs. The 5 LTRs in whom anti-RBD antibodies were not induced after the second vaccination all showed induction of the antibodies after the third vaccination. However, there was one individual who, despite having a positive

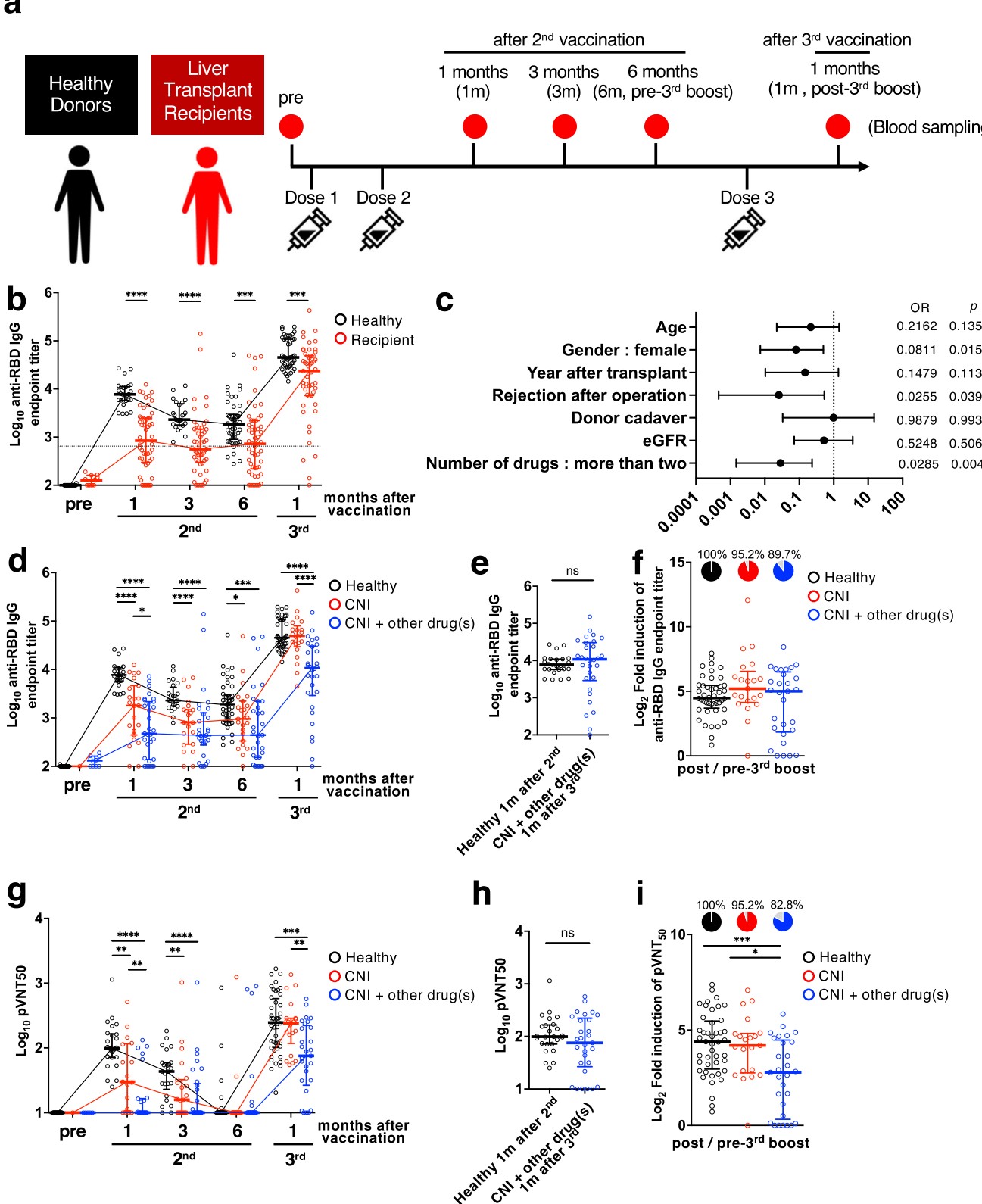

plasma anti-RBD antibody titer after the second dose, did not benefit from the third booster dose and tested negative. This individual was taking three medications, namely CNI, MMF, and steroids (5 mg/day), and had a low anti-RBD antibody titer even after the second vaccination.

Additionally, LTRs were regrouped based on clinical information apart from medication (Supplementary Fig. 1). Antibody titers were considerably

lower in deceased donor liver transplant (DDLT) than in living-donor liver transplant (LDLT) (Supplementary Fig. 1a). Furthermore, antibody titers in LTRs less than 12 years after transplantation were lower (Supplementary Fig. 1b). LTRs who experienced rejection reactions after transplantation also exhibited lower antibody titers than those who did not (Supplementary Fig. 1c). LTRs who have taken MMF also exhibited lower antibody titers

**Fig. 1 | Changes in anti-RBD IgG titers and plasma neutralizing activity in HDs and LTRs immunized by COVID-19 mRNA vaccine. a** Schematic overview of the cohort. **b** Anti-RBD IgG endpoint titers in HDs (black) and LTRs (red) (sample size, pre: 25 vs 12, 1 m after 2nd: 25 vs 54, 3 m after 2nd: 24 vs 53, 6 m after 2nd: 44 vs 54, 1 m after 3rd: 44 vs 51). **c** Multivariable logistic regression model (OR and 95% CI) for predictors of weak and strong responders (lower and higher than median antibody titer in HDs at 1 month after third vaccination, respectively). **d** Anti-RBD antibody titers in HDs (black), LTRs taking only a calcineurin inhibitor (CNI group, red) and LTRs taking CNI and other medications (CNI+other drug(s) group, blue) (sample size, pre: 25 vs 1 vs 11, 1 m after 2nd: 25 vs 20 vs 27, 3 m after 2nd: 25 vs 23 vs 30, 6 m after 2nd: 44 vs 23 vs 31, 1 m after 3rd: 44 vs 21 vs 29). **e** Anti-RBD IgG endpoint titers in HDs 1 month after 2nd vaccination (black) and in CNI+other drug(s) group 1 month after 3rd vaccination (blue) (sample size, 25 vs 29). **f** Fold-induction in anti-RBD IgG endpoint titers after third vaccination (HDs: black, CNI: red, CNI+other drug(s): blue). Pie charts represent the proportion of individuals with fold-induction > 1, and gray slice shows frequency of negative responders. (sample size, 44 vs 21 vs 29). **g** pVNT$_{50}$ against SARS-CoV-2 Wuhan-1 (HDs: black, CNI: red, CNI+other drug(s): blue). **h** pVNT$_{50}$ in HDs 1 month after 2nd vaccination (black) and in CNI+other drug(s) group 1 month after 3rd vaccination (blue) (sample size, 25 vs 29). **i** Fold-induction in pVNT$_{50}$ after third vaccination. Pie charts represent the proportion of individuals with fold-induction > 1, and gray slice shows frequency of negative responders (HDs: black, CNI: red, CNI+other drug(s): blue) (sample size, 44 vs 21 vs 29). *P* values (two-sided) were calculated using the Mann–Whitney *U*-test. All experiments were performed once. Error bars indicate the interquartile range.

than those who have not (Supplementary Fig. 1d). These factors are related to the regimen of immunosuppressive drugs, and the multivariate analysis suggested that the number of drugs has the most significant impact. Noteworthily, antibody titers of 89.7% in the CNI+other drug(s) group were increased by the third vaccination, and the fold induction of antibody titers in the CNI+other drug(s) group was similar to that in HDs (Fig. 1f; *p* = 0.7666).

Next, we measured the changes in the neutralizing activity of plasma from HDs and LTRs (Fig. 1g). Neutralizing activity in most of the CNI +other drug(s) was below the detection limit after the second vaccination, and was significantly lower than that in HDs after the third vaccination (*p* = 0.0001). Contrarily, the neutralizing activity in the CNI+other drug(s) group after the third vaccination was similar to that in HDs one month after the second vaccination (Fig. 1h; *p* = 0.2985). Furthermore, although the fold-induction of neutralizing activity in CNI+other drug(s) by the third vaccination was significantly lower than that of HDs, 82.8% of the CNI +other drug(s) group got a booster effect (Fig. 1i) (*p* = 0.0006 among HDs vs. CNI+other drug(s)). These results suggest that the third doses of mRNA vaccine are worthwhile for the induction of neutralizing activity in LTRs, but may not be sufficient compared to HDs.

## CD4$^+$ T-cell responses correlate with anti-RBD IgG titers in HDs and LTRs
Generally, immunosuppressive drugs, including CNIs, contribute to the suppression of T-cell responses. To investigate whether the reduction in antibody titers in LTRs is affected by changes in CD4 helper T-cell function, we performed flow cytometry analysis to evaluate the CD4$^+$ T-cell responses. The frequency of total SARS-CoV-2 spike-specific CD4$^+$ T cells was measured using CD154 as an activation marker (Supplementary Fig. 2a). The frequency of spike-specific CD4$^+$ T cells in CNI+other drug(s) at 1, 3, and 6 months after the second vaccination was significantly lower compared to HDs (Fig. 2a; *p* = 0.0117, *p* = 0.0208, and *p* = 0.0047 at 1, 3, and 6 months after the second vaccination, respectively). There was no significant difference between HDs and the CNI group at 1 month (*p* > 0.9999), 3 months (*p* = 0.6506), and 6 months (*p* = 0.1379) after the second vaccination. Moreover, there were significant differences between the CNI and CNI+other drug(s) groups 3 months (*p* = 0.024), and 6 months (*p* = 0.0051) after the second vaccination (Fig. 2a). However, there is no significant difference among HDs, the CNI group, and the CNI+other drug(s) group after the third vaccination. Regardless of HDs or LTRs, spike-specific CD4$^+$ T cells decreased over time after the second mRNA vaccination (Supplementary Fig. 2b).

Next, we measured the cytokine profiles of the total spike-specific CD4$^+$ T cells (Fig. 2b, c, Supplementary Fig. 2a). The frequency of Th1 cells in CNI+other drug(s) after the second vaccination was significantly lower compared to HDs (Fig. 2b). On the contrary, the frequency of Th2 cells was higher in the CNI group than in HDs (Fig. 2c). The frequency of total CD154$^+$ spike-specific CD4$^+$ T cells and Th1 cells increased by the third mRNA vaccination in HDs and LTRs, and there was no significant difference between HDs and LTRs after the third vaccination (Fig. 2a, b).

We next examined the effect of the third booster on memory CD4$^+$ T cell responses by calculating the fold-induction of CD154$^+$, Th1, and Th2 cell frequencies. We observed a boost effect in ~75% of individuals for CD154$^+$ and Th1 cells in all groups, and in ~50% of individuals for Th2 cells (Fig. 2d–f). Furthermore, Th1/Th2 ratio in LTRs was significantly lower compared to HDs (Fig. 2g), suggesting that LTRs are more susceptible to the induction of Th2-biased CD4$^+$ T-cell responses.

We next evaluated the correlation between CD4$^+$ T-cell and antibody responses. One month after the second vaccination, the frequency of CD154$^+$CD4$^+$ T and Th1 cells was positively correlated with anti-RBD antibody titers in HDs and LTRs (Fig. 2h). Moreover, CD4$^+$ T-cell frequency before the third vaccination positively correlated with antibody titers after the third vaccination (HDs: *r* = 0.299, *p* = 0.049 for CD154$^+$CD4$^+$ T cells vs. anti-RBD IgG; LTRs: *r* = 0.483, *p* = 0.0004 for CD154$^+$CD4$^+$ T cells vs. anti-RBD IgG; *r* = 0.433, *p* = 0.0019 for Th1 CD4 T cells vs. anti-RBD IgG). These results suggest that long-term CD4$^+$ T-cell responses after the second vaccination contribute to the booster effect on antibody levels after the third vaccination.

## Characterization of spike-specific CD8$^+$ T-cell responses in HDs and LTRs
In addition to antibodies and CD4$^+$ T cells, CD8$^+$ T-cell responses also contribute to defense against SARS-CoV-2 infection[22,23]. However, COVID-19 mRNA vaccines reportedly have a lower ability to induce CD8$^+$ T cells than CD4$^+$ T cells[24]. Moreover, few reports demonstrate CD8$^+$ T-cell responses to mRNA vaccines in LTRs. Therefore, we investigated whether spike-specific CD8$^+$ T cells were induced in LTRs and compared their frequency with HDs. We defined 4-1BB$^+$CD69$^+$CD8$^+$ T cells as spike-specific CD8$^+$ T cells in the PBMCs stimulated with spike peptides (Supplementary Fig. 3a). Spike-specific CD8$^+$ T cells were detected in 100% of HDs and 93% of LTRs 1 month after the second vaccination (Fig. 3a). However, the frequency of spike-specific CD8$^+$ T cells by the third vaccination did not increase in most HDs and LTRs (Fig. 3b, HDs 55.8%, CNI 55%, and CNI+other drug(s) 42.9%). Compared to HDs, the frequency of LTRs was significantly lower at all time points, regardless of taking single or multiple drugs (Fig. 3a). Furthermore, in contrast to antibody responses, there was no correlation between spike-specific CD8$^+$ and CD4$^+$ T cell responses (Fig. 3c). These results suggest that the third boost effect on memory T-cell responses differs between CD4$^+$ and CD8$^+$ T cells. We then checked the differentiation status of the spike-specific CD8$^+$ T cells induced by vaccination using CD27, CD45RO, and CD57 markers to define central memory (CM; CD27$^+$CD45RO$^+$), effector memory (EM; CD27$^-$CD57$^-$), and effector (CD27$^-$CD57$^+$) subsets. As a result, the phenotypes of spike-specific CD8$^+$ T cells were changed from CM to EM at 6 months after 2nd vaccination in both the HDs and LTRs who showed positive effects of boosting spike-specific CD8$^+$ T-cell responses (Healthy boost+ and LTR boost + ), although the phenotypes of total memory CD8$^+$ T cells were not changed over time (Fig. 3d, e). After 3rd mRNA vaccination, HDs and LTRs showed different phenotypes of spike-specific CD8$^+$ T cells, with decreased CM and increased EM and Effector in HDs, but a trend toward increased CM in LTRs.

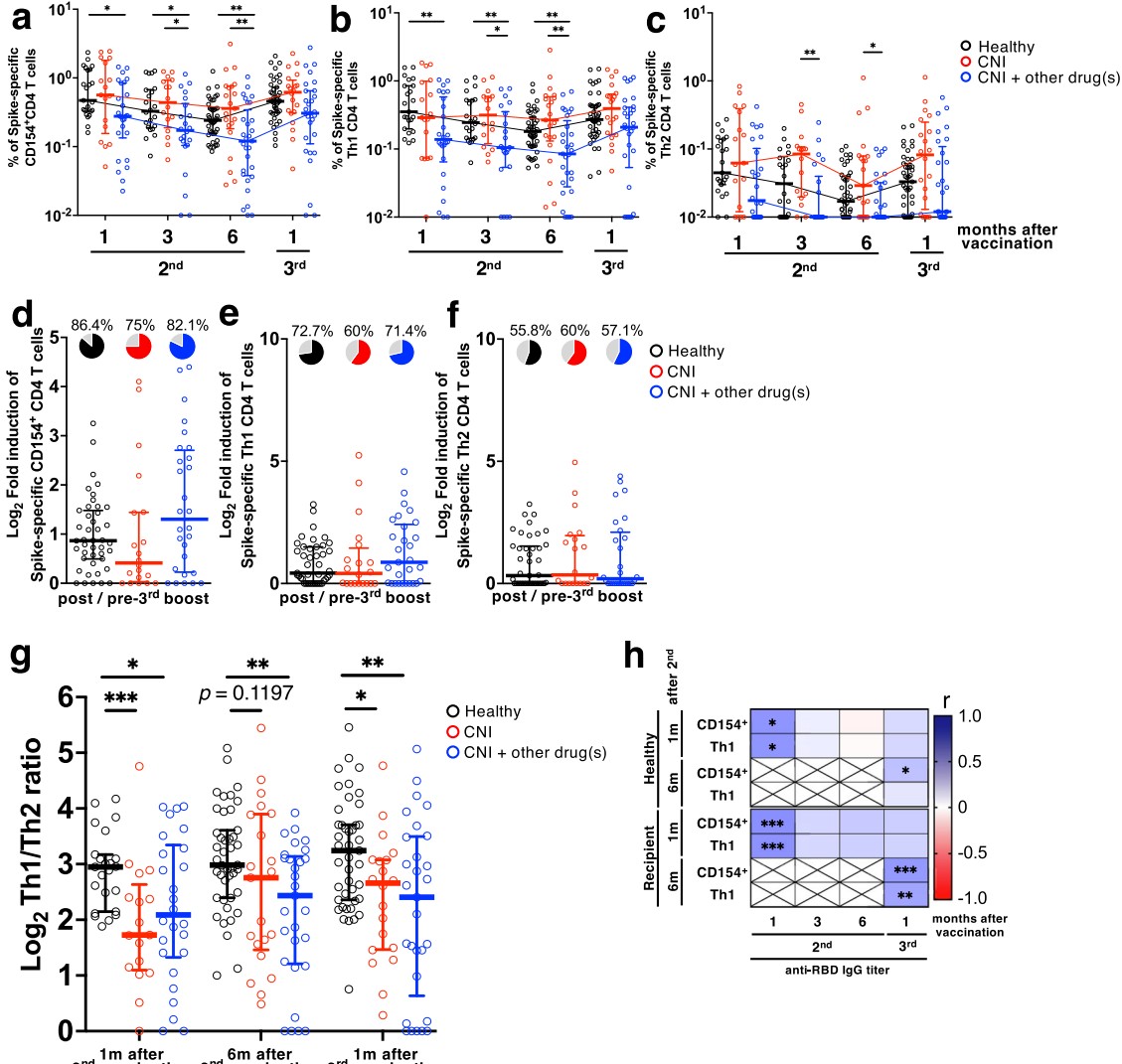

**Fig. 2 | CD4+ T-cell responses correlate with anti-RBD IgG titers in HDs and LTRs.** Frequency of spike-specific CD154+ (**a**), Th1 (**b**), and Th2 (**c**) CD4+ T cells in total memory T cells from HDs (black), CNI group (red), and CNI+other drug(s) group (blue). **d–f** Fold-induction of spike-specific CD154+, Th1, and Th2 CD4+ T cells by the third vaccination. Pie charts represent the proportion of individuals with fold-induction higher than 1, and gray slice shows frequency of negative responders. (HDs: black, CNI: red, CNI+other drug(s): blue). **g** The ratio of spike-specific Th1 to Th2 CD4+ T cells (HDs: black, CNI: red, CNI+other drug(s): blue). P

values (two-sided) in (**a**) to (**g**) were calculated using the Mann-Whitney U-test. **h** Correlation matrix of antibody and CD4+ T-cell responses in HDs and LTRs. Shades of blue represent positive correlations approaching 1, while shades of red denote negative correlations nearing -1. P values (two-sided) were calculated using the Spearman's rank test. Sample size, 1 m after 2nd: 23 vs 17 vs 26, 3 m after 2nd: 22 vs 16 vs 22, 6 m after 2nd: 43 vs 21 vs 29, 1 m after 3rd: 43 vs 20 vs 29. All experiments were performed once. Error bars indicate the interquartile range.

Furthermore, we previously reported that differences in the expression patterns of cytotoxic molecules could observe qualitative differences in mRNA vaccine-induced spike-specific CD8+ T cells[20]. Therefore, we compared the expression of cytotoxic molecules in spike-specific CD8+ T cells between HDs and LTRs. Supplementary Fig. 3b shows the expression patterns of GZMA, GZMB, and Perforin, and gating. Regardless of HDs or LTRs, most spike-specific CD8+ T cells expressed GZMA before and after the third vaccination (Fig. 3f). The proportion of cells expressing GZMA in CNI+other drug(s) was significantly, but slightly, lower than that in HDs before the third vaccination (p = 0.0237). However, the proportion of cells expressing GZMB and Perforin was not different between HDs and LTRs before and after the third boost (Fig. 3g, h). Furthermore, the expression profiles of GZMA, GZMB, and Perforin were not significantly different between the groups (Supplementary Fig. 4a, b). The proportion of subpopulations expressing GZMA, GZMB, and Perforin was approximately 20% in the spike-specific CD8+ T cells of each group, and the proportion of subpopulations expressing only GZMA was over 50% (Fig. 3i). However, we

did not observe any qualitative differences in spike-specific CD8+ T cells induced by the third boost.

## Antibody against SARS-CoV-2 variants of concern induced by mRNA vaccine

HDs and LTRs were vaccinated with an mRNA vaccine based on the Wuhan-1 strain, and the induced antibodies potentially reduced the effectiveness against the recently emerged Omicron sublineages. Therefore, we measured the antibody titers before and after the third boost against RBD corresponding to the Omicron sublineages, and found that anti-RBD antibody titers before the third boost against all sublineages were significantly reduced compared to those against the Wuhan-1 (Fig. 4a, b). Among sublineages, the anti-RBD antibody titers against BQ.1.1 and XBB were particularly reduced (HDs, 8.43-fold reduction; CNI, 5.23-fold reduction; CNI+other drug(s), 4.41-fold reduction against BQ.1.1, HDs, 11.9-fold reduction; CNI, 6.35-fold reduction; CNI+other drug(s), 4.41-fold reduction against XBB). Furthermore, the neutralizing activity before the

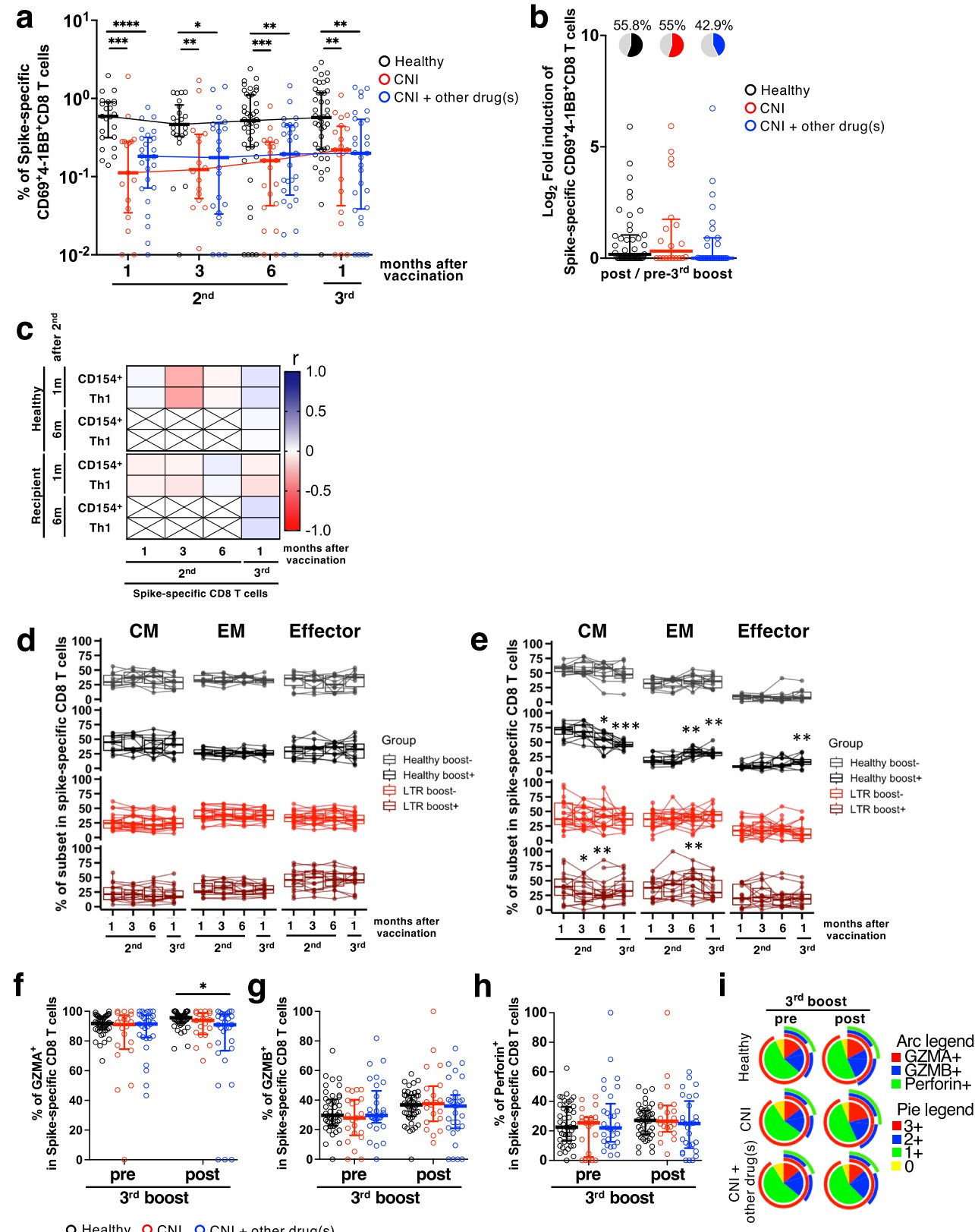

third boost was below the detection limit for BA.5, BQ.1.1, and XBB in most individuals (Fig. 4c). Furthermore, there was no change in the trend toward lower antibody titers for each Omicron sublineage (Fig. 4d, e). In particular, the CNI+other drug(s) group showed significantly lower anti-RBD antibody levels against all sublineages than the HDs and CNI groups.

Additionally, there was a slight improvement in neutralizing activity against the BA.5 strain, but not BQ.1.1 and XBB strains, by the third vaccination (Fig. 4f).

Collectively, these results suggest that the third vaccination with the Wuhan-1 mRNA vaccine may not be sufficient to induce antibody

**Fig. 3 | CD8$^+$ T-cell responses were reduced in LTRs but there no significant difference in the expression of cytotoxic molecules. a** Frequencies of spike-specific CD69$^+$4-1BB$^+$CD8$^+$ T cells in total memory T cells from HDs (black), CNI group (red), and CNI+other drug(s) group (blue). **b** Fold-induction of spike-specific CD69$^+$4-1BB$^+$CD8$^+$ T cells after third vaccination. Pie charts represent the proportion of individuals with fold-induction > 1, and gray slice shows frequency of negative responders (HDs: black, CNI: red, CNI+other drug(s): blue). **c** Correlation matrix of CD4$^+$ and CD8$^+$ T-cell responses. Shades of blue represent positive correlations approaching 1, while shades of red denote negative correlations nearing -1. *P* values were calculated using the Spearman's rank test. Frequencies of CM, EM and effector within CD8$^+$ total memory T cells (**d**) and spike-specific CD69$^+$4-1BB$^+$ CD8$^+$ T cells (**e**) in individuals who did (boost + ) or did not (boost-) receive boost effect from 3 doses of mRNA vaccine (HDs boost–: gray, HDs boost + : black, LTRs

boost–: red, LTRs boost + : dark red). *P* values (two-sided) were calculated using the Wilcoxon matched-pairs signed rank test compared to 1 month after 2$^{nd}$ vaccination. Frequency of spike-specific CD69$^+$4-1BB$^+$CD8$^+$ T cells expressing GZMA (**f**), GZMB (**g**), and Perforin (**h**) (HDs: black, CNI: red, CNI+other drug(s): blue). **i** Expression of multiple cytotoxic molecules in spike-specific CD69$^+$4-1BB$^+$CD8$^+$ T cells. Each color's arc length and pie chart's area represent the expression of each cytotoxic molecule (GZMA: red, GZMB: blue, Perforin: green) and cells expressing the indicated number of cytotoxic molecules (0: yellow, 1: green, 2: blue, 3: red), respectively. *P* values (two-sided) in (**a**), (**b**), (**f**), (**g**), and (**h**) were calculated using the Mann–Whitney *U*-test. Sample size, 1 m after 2nd: 23 vs 17 vs 26, 3 m after 2nd: 22 vs 16 vs 22, 6 m after 2nd: 43 vs 21 vs 29, 1 m after 3rd: 43 vs 20 vs 29. All experiments were performed once. Error bars indicate the interquartile range.

responses against Omicron sublineages, particularly BQ.1.1 and XBB, in HDs and LTRs.

## Cellular immune responses against SARS-CoV-2 variants of concern induced by mRNA vaccine

Finally, we investigated the differences in cellular immunity against Omicron sublineages between HDs and LTRs. The frequency of spike-specific CD154$^+$CD4$^+$ T cells was evaluated in PBMCs before the third boost. There was no difference in response to the Wuhan-1 and mutant strains in all groups (Supplementary Fig. 5a, b). The same trend was observed for spike-specific Th1 CD4$^+$ T cells (Fig. 5a, b). However, the frequency of CD154$^+$CD4$^+$ T cells and Th1 cells responding to mutant strains in HDs after the third boost was significantly and slightly lower than that of cells responding to Wuhan-1(Supplementary Fig. 5c, d, Fig. 5c, d). The same trend was observed in spike-specific Th2 CD4$^+$ T cells (Supplementary Fig. 4e–h). These results indicate that, unlike antibody responses, CD4$^+$ T-cell responses induced by mRNA vaccines can react to Omicron sublineages. Moreover, LTRs resulted in CD4$^+$ T-cell responses to Omicron sublineages with comparable reactivity to those in HDs.

Next, we investigated CD8$^+$ T cell responses to Omicron sublineages. Interestingly, the frequency of spike-specific CD8$^+$ T-cell responses to mutant strains was not significantly decreased, regardless of the pre- and post-third boost (Fig. 6a, b). The fold-changes in the frequency of CD8$^+$ T-cell responses to mutant strains relative to Wuhan-1 are shown (Fig. 6c, d). Collectively, these results demonstrate that mRNA vaccines induce CD8$^+$ T-cell responses reactive to BA.5, BQ.1.1, and XBB mutant strains and that these responses are maintained in LTRs.

## Discussion

In this study, the anti-RBD IgG titers in LTRs induced by the mRNA vaccines were lower than those in HDs after the second and third vaccination. Multivariate analysis based on LTRs' background information revealed that the use of multiple immunosuppressive drugs was a key factor in the lack of antibody induction, which is consistent with recent studies[7,8]. Even in LTRs receiving multiple medications, the third mRNA vaccination induced antibody responses similar to those observed in HDs after the second vaccination. However, neutralizing antibodies obtained with only three doses of the Wuhan-1-type mRNA vaccine are insufficient against Omicron sublineages. It could be necessary to receive a fourth dose of a vaccine or a bivalent vaccine[15,25], especially for LTRs.

Furthermore, in this cohort, eight patients had a history of rituximab treatment, which targets B cells. A study in patients with multiple sclerosis receiving anti-CD20 therapy within 20 weeks before mRNA vaccination showed that antibody production following mRNA vaccination was drastically reduced, while T-cell responses were induced[26]. However, in our cohort, no LTRs received anti-CD20 therapy within 1 year before vaccination, and rituximab treatment did not affect antibody induction. Another critical point is that, unlike in Western countries, LDLT is the primary method in our cohort. Our study could uniquely compare LDLT with DDLT (LDLT; *n* = 46, DDLT; *n* = 8) and evaluate the changes in immune responses over time in vaccine efficacy in LTRs from LDLT. Furthermore,

passive immunotherapy such as HBIG is mainly used in LTR because HBV vaccine is less effective in LTRs[27]. While the HBV vaccine is a recombinant protein vaccine, the mRNA vaccine developed as a new modality during the SARS-CoV-2 pandemic is a potentially effective platform for inducing neutralizing antibodies even in immunosuppressed LTR.

CNIs are the most commonly used immunosuppressive drugs targeting T cells in LTRs. CNIs inhibit calcineurin, resulting in the inactivation of the nuclear factor of activated T-lymphocytes (NFAT) and suppression of IL-2 production[28]. In other words, CNIs specifically target T cells, but for the evaluation of mRNA vaccine effects in LTR, the focus is on humoral immunity due to technical limitations. Notably, the early induction of CD4$^+$ T-cell responses by mRNA vaccines is reportedly necessary for antibody production[29]. Therefore, it is crucial to investigate the effectiveness of mRNA vaccine-induced T-cell responses in LTRs. Our findings demonstrate that CD4$^+$ T-cell responses before the third mRNA vaccination significantly correlated with anti-RBD IgG titers after the third vaccination. These results suggest that the long-term maintenance of CD4$^+$ T cell responses is an important factor for the acquisition of high antibody titers in LTRs.

Furthermore, taking multiple drugs reduced spike-specific CD4$^+$ T-cell responses after the second vaccination, similar to antibody titers; however, the third vaccination significantly improved CD4$^+$ T-cell responses. Therefore, a third mRNA vaccination is considered effective for acquiring immune responses in LTRs regarding both antibody and CD4$^+$ T-cell responses. Additionally, CNI alone did not affect the induction of spike-specific CD154$^+$CD4$^+$ T cells. However, CD4$^+$ T cells were biased toward the Th2 phenotype in the LTRs. The induced Th2-biased CD4$^+$ T cells in mice and hamsters reportedly lead to vaccine-associated enhanced respiratory disease (VAERD) upon SARS-CoV-2 infection[30,31]. Although the occurrence of VAERD caused by COVID-19 mRNA vaccines in humans has not been verified, it is necessary to consider the possibility of VAERD in LTRs. Moreover, although COVID-19 mRNA vaccines have a strong Th1 induction ability[32,33], it is possible that the environment in which CD4 naive T cells are more prone to differentiate into Th2 phenotype is created due to the effect of CNI. It has been shown that CD4 naive T cells polarize into Th1 phenotype upon receiving strong TCR signals and into Th2 phenotype upon receiving weak TCR signals in mice[34]. Furthermore, previous studies have demonstrated a clear shift from Th1 to a Th2 cytokine-secreting profile as an additional mechanism of immunosuppression by cyclosporin A[35] and steroids[36]. However, Th1/Th2 ratio in LTRs was not different after the second and third mRNA vaccination, suggesting that the possibility of Th2-biased reactions becoming predominant after multiple mRNA vaccinations is low.

The cytolytic activity of spike-specific CD8$^+$ T-cell responses is a key factor for reducing the risks of severity against SARS-CoV-2 infection in HDs[37]. Recent studies suggest that mRNA vaccines induce weaker CD8$^+$ T-cell responses in healthy donors compared to CD4$^+$ T-cell responses[38,39]. In solid organ transplant recipients, including liver and kidney transplantation, the frequency of spike-specific CD8$^+$ T cells was found to be significantly lower after 2nd mRNA vaccination compared to healthy donors[9]. This finding also indicates that solid organ transplant recipients have weaker CD4$^+$ and CD8$^+$ T-cell responses compared to other groups of

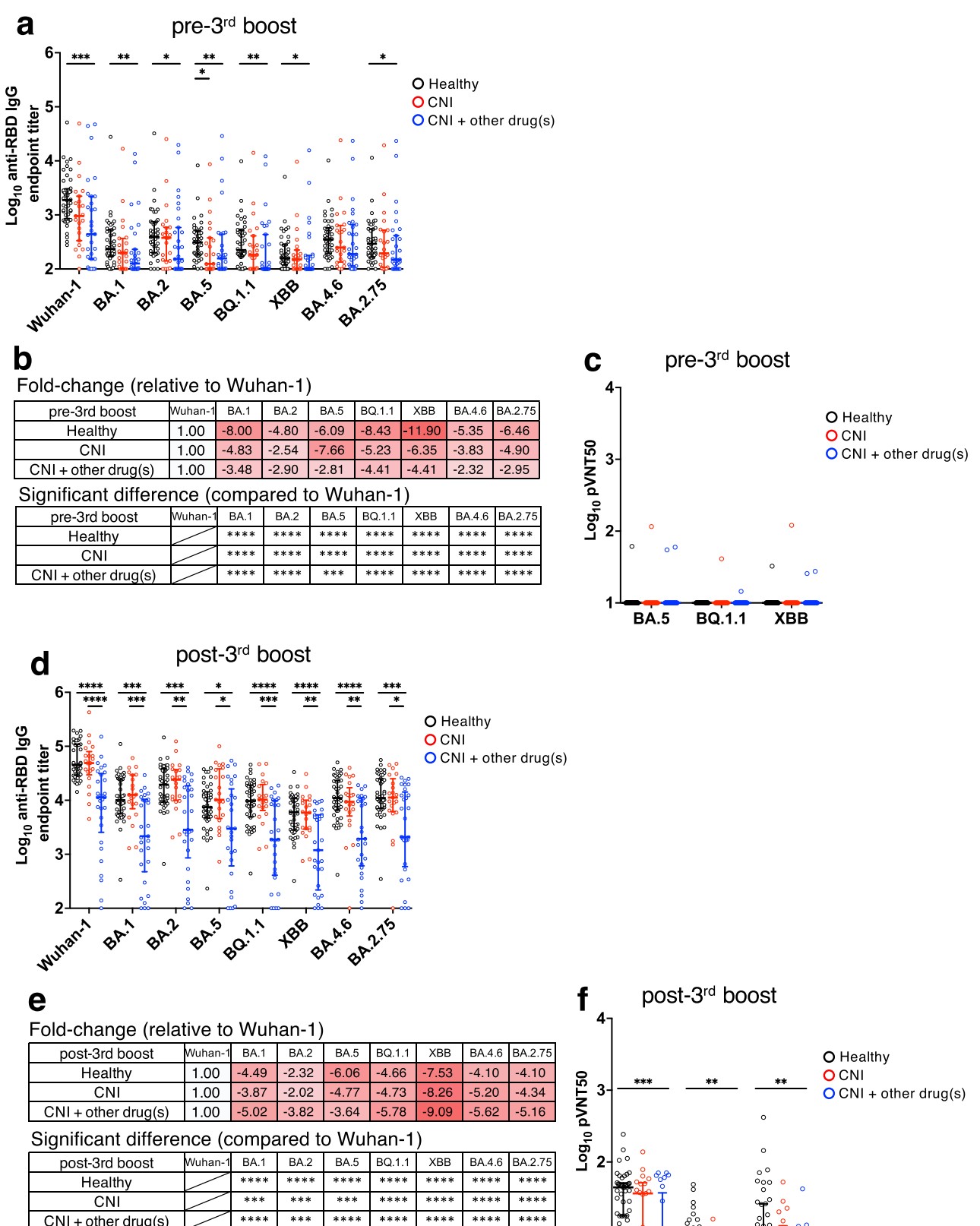

**Fig. 4 | Antibody against SARS-CoV-2 variants of concern induced by mRNA vaccine. a, d** Anti-RBD antibody endpoint titers against indicated strains at (**a**) pre- and (**d**) post-third boost (HDs: black, CNI: red, CNI+other drug(s): blue). Fold-change of anti-RBD IgG against variants of concern endpoint titers at (**b**) pre- and (**e**) post-third boost relative to Wuhan-1. The minus symbol denotes increased resistance. Shades of red indicate a decrease in antibody titers, with darker shades signifying a larger negative fold change. pVNT$_{50}$ against strains at (**c**) pre- and (**f**) post-third boost (HDs: black, CNI: red, CNI+other drug(s): blue). $P$ values (two-sided) in (**a**), (**c**), (**d**), and (**f**) were calculated using the Mann–Whitney $U$-test. $P$ values (two-sided) in (**b**) and (**e**) were calculated using the Wilcoxon matched-pairs signed rank test. Sample size, pre-3rd boost: 44 vs 23 vs 31, post-3rd boost: 44 vs 21 vs 30). All experiments were performed once. Error bars indicate the interquartile range.

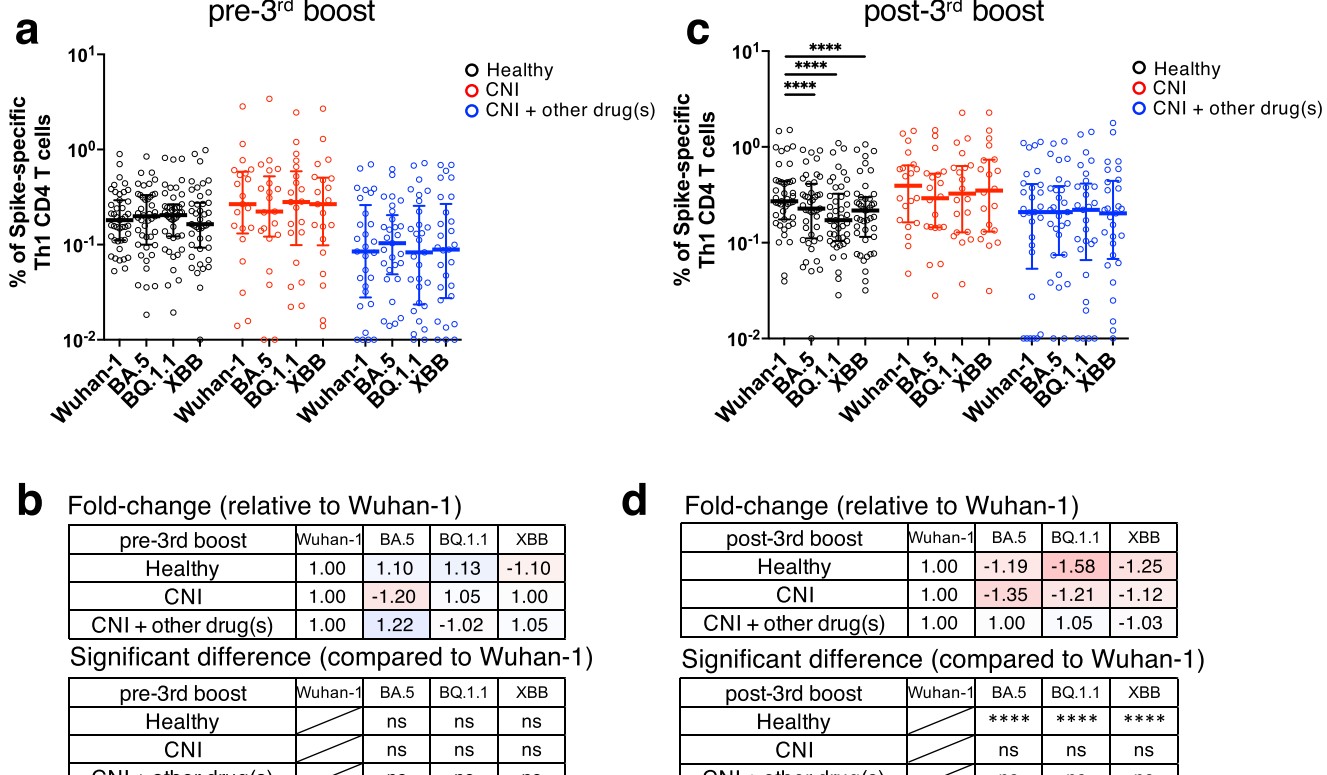

**Fig. 5 | CD4⁺ T-cell responses against SARS-CoV-2 variants of concern induced by mRNA vaccine. a** Comparison of spike-specific Th1 CD4⁺ T-cell frequency against spike peptides in CD4⁺ total memory T cells at pre-third boost (HDs: black, CNI: red, CNI+other drug(s): blue). **b** Fold-change of spike-specific Th1 CD4⁺ T-cell frequency against variants of concern at pre-third boost relative to Wuhan-1. The minus symbol denotes increased resistance. Shades of blue represent an increase in fold change, with darker shades indicating a larger positive fold change. Conversely, shades of red denote a decrease, with darker shades signifying a larger negative fold change. **c** Comparison of spike-specific Th1 CD4⁺ T-cell frequency against spike peptides in CD4⁺ total memory T cells at post-third boost (HDs: black,

CNI: red, CNI+other drug(s): blue). **d** Fold-change of spike-specific Th1 CD4⁺ T-cell frequency against variants of concern at post-third boost relative to Wuhan-1. The minus symbol denotes increased resistance. Shades of blue represent an increase in fold change, with darker shades indicating a larger positive fold change. Conversely, shades of red denote a decrease, with darker shades signifying a larger negative fold change. *P* values (two-sided) were calculated using the Wilcoxon matched-pairs signed rank test. Sample size, 1 m after 2nd: 23 vs 17 vs 26, 3 m after 2nd: 22 vs 16 vs 22, 6 m after 2nd: 43 vs 21 vs 29, 1 m after 3rd: 43 vs 20 vs 29. All experiments were performed once. Error bars indicate the interquartile range.

immunocompromised patients, such as primary immunodeficiency syndromes, AIDS, hematopoietic stem cell transplant recipients, and chronic lymphocytic leukemia patients. In our study, the frequency of spike-specific CD8⁺ T cells was significantly lower than that of healthy donors at all time points. Although spike-specific CD8⁺ T-cell responses decreased quantitatively in CNI group and CNI+other drug(s) group compared to healthy donors, there is no significant difference between healthy donors and LTR in terms of the booster effect of mRNA vaccines (Fig. 3b).

In recent years, it has been reported that the expression of GZMB and perforin in bulk CD8⁺ T cells, not antigen-specific CD8⁺ T cells, in SARS-CoV-2 infected patients is higher than that in healthy donors[40]. This suggests that the induction of CD8⁺ T cells expressing high levels of cytotoxic molecules contributes potentially to the suppression of COVID-19 severity. However, a recent study using MHC multimers has demonstrated that the expression levels of GZMB and perforin in SARS-CoV-2 spike epitope-specific CD8⁺ T cells decrease over time after 2nd mRNA vaccination, and do not increase after 3rd vaccination[41]. Therefore, the ability of mRNA vaccines to induce CD8⁺ T cells with high cytotoxic activity is limited. In our study, the subpopulations expressing GZMA, GZMB, and perforin were not different between HDs and LTRs, indicating that mRNA vaccine-induced spike-specific CD8⁺ T cells are quantitatively, but not qualitatively, limited to LTRs. In summary, no improvement was observed in both the quantitative or qualitative aspects of spike-specific CD8⁺ T cells even after 3rd vaccination in both healthy donors and LTR in our study. Moreover, both CD4⁺ and CD8⁺ T-cell responses induced by Wuhan-1 mRNA vaccines were reactive to Omicron

sublineages BA.5, BQ.1.1, and XBB in HDs and LTRs, suggesting that inducing T-cell responses is crucial for dealing with new mutant strains.

We also examined the relationship between LTRs' background information and the induction of spike-specific CD8⁺ T cells (Fig. 1c). Although taking multiple drugs did not affect the responsiveness of spike-specific CD8⁺ T cells to the third boost, unlike the antibody responses, a decline in eGFR affected CD8⁺ T-cell responses (Supplementary Fig. 6A). This result suggests that LTRs with decreased kidney function may be a potential risk factor for weaker CD8⁺ T-cell responses, as shown by an epidemiological study demonstrating high serum creatinine levels in severe COVID-19 patients[4]. Since CD8⁺ T-cell responses during SARS-CoV-2 infection in LTRs with decreased kidney function have not been evaluated, further investigation is needed to address this. Furthermore, the median steroid dose administered to the liver transplant recipients in this study was 5 mg/day (range: 0.5–10 mg/day). Within this range, an effect was observed on the antibody titer, but no effect on the T-cell response was noted (Supplementary Fig. 7a).

Finally, the booster effect of the third vaccination was found in terms of spike-specific CD4⁺ T-cell and antibody responses but not CD8⁺ T-cell responses in HDs and LTRs. Regarding the quantitative changes in CD8⁺ T-cell responses, only ~50% of HDs and LTRs obtained a boosting effect by the third vaccination (Fig. 3b). This cannot be explained simply by HDs vs. LTRs. In our phenotypic analysis of spike-specific CD8 T cells, we found that the differentiation/maturation of the spike-specific CD8⁺ T cells after two doses of mRNA vaccine could be a key factor for boosting spike-specific CD8⁺ T cells by the third vaccination. Further investigation is still required

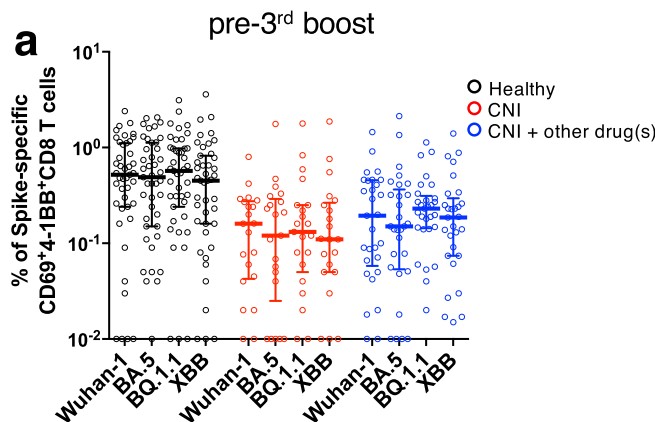

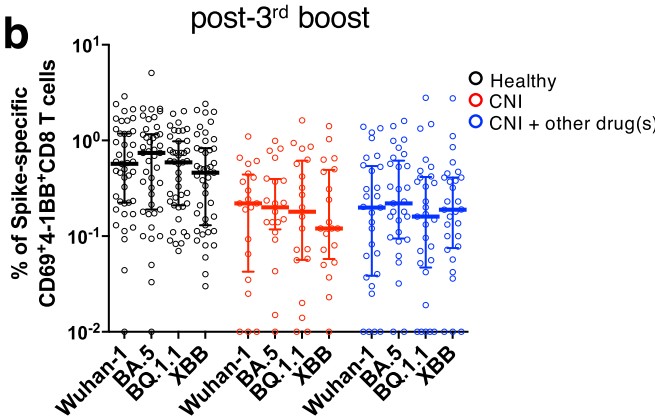

**c** Fold-change (relative to Wuhan-1)

| pre-3rd boost | Wuhan-1 | BA.5 | BQ.1.1 | XBB |
|---|---|---|---|---|
| Healthy | 1.00 | -1.06 | 1.10 | -1.16 |
| CNI | 1.00 | -1.33 | -1.22 | -1.45 |
| CNI + other drug(s) | 1.00 | -1.29 | 1.19 | -1.04 |

Significant difference (compared to Wuhan-1)

| pre-3rd boost | Wuhan-1 | BA.5 | BQ.1.1 | XBB |
|---|---|---|---|---|
| Healthy | | ns | ns | ns |
| CNI | | ns | ns | ns |
| CNI + other drug(s) | | ns | ns | ns |

**d** Fold-change (relative to Wuhan-1)

| post-3rd boost | Wuhan-1 | BA.5 | BQ.1.1 | XBB |
|---|---|---|---|---|
| Healthy | 1.00 | 1.34 | 1.08 | -1.20 |
| CNI | 1.00 | -1.10 | -1.22 | -1.83 |
| CNI + other drug(s) | 1.00 | 1.11 | -1.24 | -1.05 |

Significant difference (compared to Wuhan-1)

| post-3rd boost | Wuhan-1 | BA.5 | BQ.1.1 | XBB |
|---|---|---|---|---|
| Healthy | | ns | ns | ns |
| CNI | | ns | ns | ns |
| CNI + other drug(s) | | ns | ns | ns |

**Fig. 6 | CD8+ T-cell responses against SARS-CoV-2 variants of concern induced by mRNA vaccine.** Comparison of spike-specific CD69+4-1BB+ CD8+ T-cell frequency against spike peptides in CD8+ total memory T cells at (**a**) pre- and (**b**) post-third boost (HDs: black, CNI: red, CNI+other drug(s): blue). Fold-change of spike-specific CD8+ T-cell frequency against variants of concern at (**c**) pre- and (**d**) post-third boost relative to Wuhan-1. The minus symbol denotes increased resistance. Shades of blue represent an increase in fold change, with darker shades indicating a larger positive fold change. Conversely, shades of red denote a decrease, with darker shades signifying a larger negative fold change. *P* values (two-sided) were calculated using the Wilcoxon matched-pairs signed rank test. Sample size, 1 m after 2nd: 23 vs 17 vs 26, 3 m after 2nd: 22 vs 16 vs 22, 6 m after 2nd: 43 vs 21 vs 29, 1 m after 3rd: 43 vs 20 vs 29. All experiments were performed once. Error bars indicate the interquartile range.

to address the moleculer mechanism of this observation. While mRNA vaccines are expected to be effective platforms for various pathogens that may emerge, low CD8+ T-cell induction ability could be an issue for future mRNA vaccine development.

In summary, the third mRNA vaccination improves humoral responses and spike-specific CD4+ T-cell responses in LTRs but exhibited no booster effect for spike-specific memory CD8+ T-cell responses. Spike-specific CD4+ and CD8+ T cells can react to Omicron sublineages in HDs and LTRs, which suggests that a third mRNA vaccination could be helpful in LTRs to prevent SARS-CoV-2 infection, although the further investigation will be needed to elicit CD8 T-cell responses in not only LTRs but also HDs.

## Data availability

Source data, datasets generated and/or analyzed during the current study, are available in the paper or are appended as Supplementary Data 1. The data supporting the findings of this study are available from the corresponding author upon request.

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

## Acknowledgements

We thank all the members of the Laboratory of Precision Immunology, Center for Intractable Diseases and ImmunoGenomics, National Institutes of Biomedical Innovation, Health and Nutrition, Osaka, Japan, especially MSes. Yuki Katayama and Mami Ikeda for their excellent technical support. This study was supported by the Japan Agency for Medical Research and Development (grant numbers JP21nf0101627 and JP21fk0108493).

## Author contributions

Conceptualization, T.Nogimori, Y.N., S.K., and T.Y.; investigation, T.Nogimori, Y.N., H.M., and Y.M.; data analysis, T.Nogimori, Y.N, H.A., S.T., S.K., and T.Y.; resources, Y.N., H.M., K.S., Y.Y., Y.T., T.Noda, Y.D., H.E., and S.K.; writing - original draft, T.Nogimori and T.Y.; Writing - review and editing, all authors; funding acquisition, T.Y.; supervision, T.Y.

## Competing interests

K.S. and Y.Y. declare the following competing interests: the Research Foundation for Microbial Diseases of Osaka University. The other authors declare no competing interests.
