## [Peer Review File · Communications Medicine]

Reviewers' comments:

Reviewer #1 (Remarks to the Author):

Dear Dr. Yamamoto

Thank you for submitting your paper.

Nogimori et al. reported an prospective study on the impact of mRNA SarsCov2 vaccination in a cohort of liver transplant recipients. They assessed the humoral and cellular biological response to vaccination, after the second and third dose of vaccine.

Comments:

Elisa's assay is commercial? How was the cut-off established? Reference 20 is associated with it, I think that is not correct.

How much time passed between the blood sample of the 6 months post-2nd dose and the 3rd dose of vaccine?

What dose of steroids were the patients receiving?

How many patients had a humoral response after the 2nd dose and how many of the seronegatives were rescued with the 3rd dose?

No patient during follow-up had a SARS-CoV-2 infection?

It is not clear when was considered to have a positive CD4 or CD8 T cell response

The discussion in general and the conclusions require a deeper approach and a greater contrast with other published works.

Graphics are confusing and not self explanatory.

Reviewer #4 (Remarks to the Author):

Thank you for inviting me to review this article.

Nogimori and Nagatsuka et al. prospectively evaluated humoral and cellular immune memory response among liver transplant recipients who were booster-vaccinated with mRNA vaccines.

Comments/suggestions

Although there have been few studies in the literature, the authors of the current manuscript have performed exhaustive work on cellular and humoral response assessment and performed several sub-group analysis.

The authors assessed at 1 month after the booster dose. Immune response is expected to be high in the early period. But how long the immunity lasts after the booster is not addressed in the study, which is the actual lacunae in the literature. And how many individuals developed breakthrough infections? this has not been answered.

The time between vaccinations (1,2 and 3rd dose) is not mentioned anywhere in the manuscript. It may be possible that individuals who receive early booster (shorter gap between 2nd and 3rd dose) after 2nd dose have a high response. Or did everybody in LTR and HD receive all vaccinations at the same time?

I assume everyone was a homologous recipient of vaccination. Future studies could evaluate impact of heterologous vaccination, which has been reported to be more beneficial.

The authors report lack of CD8 response after booster.

Figure 1E/H: The authors mention that anti-RBD and pVNT50 in LTR reach comparable levels with HD after 3rd dose. (1 month after 2nd dose in the healthy group vs. 1 month after 3rd dose in LTR group.) The endpoints are different. What was the comparison at 1 month post 3rd dose among these groups? I may have missed it probably.

Line 259/260 authors claimed that "Furthermore, Th1/Th2 ratio in LTRs was significantly lower compared to HDs (Figure 2G), suggesting that LTRs are more susceptible to the induction of Th2-biased CD4+T-cell responses". In general, LTRs with immune suppressive drugs usually have Th2 lineage and more Th2 cytokines compared to Th1.

Throughout the study authors have compared all the parameters between healthy and LTRs (CNI/CNI+other drugs) at different time points. This is important although it would have been interesting to learn the comparison between two groups CNI alone (n=23) vs. CNI +other drugs (n=31). Data can be analysed within the same group at different time (5 time points) points with statistical analysis. Also, mention this n of each group somewhere in the results. i.e., 23 vs. 31 vs. 44. Similarly, I don't see a number (sample size) anywhere for all sub-group analysis. For example: comparing LDLT vs. DDLT; recipients 12 years after and less. Why this 12 years was chosen?

MMF and mTORi have been reported to affect the immune response and it is suggested to stop this drug in infected individuals (PMID: 32750442; PMID: 36374707). Although this hypothesis is contentious it would be interesting to know if there was any differential expression of immune response in those on MMF/mTORi.

Did you analyse Th1/Th2 ratio between LTR and HD pre and post which would have added more value to the current study.

CD4 and CD8 naïve cells in the marrow of LTR are unaffected, while the CD4 effector memory and CD4 and CD8 effector cells are lower in LTRs which has been reported in the literature. Similarly, the authors report that after booster dose CM changes to EM. However, on Page no 24, Figure 3, data for CD8 memory cells of 3rd post 1-month vaccination is missing both. Did the authors perform the analysis or missed?

There are several panels (A-H). Please number them sequentially in the manuscript in bold letters.

Line 289: over- r is missing

Reviewers' comments:

Reviewer #1 (Remarks to the Author):

Dear Dr. Yamamoto

Thank you for submitting your paper.

Nogimori et al. reported an prospective study on the impact of mRNA SarsCov2 vaccination in a cohort of liver transplant recipients. They assessed the humoral and cellular biological response to vaccination, after the second and third dose of vaccine.

Comments:

Elisa's assay is commercial? How was the cut-off established? Reference 20 is associated with it, I think that is not correct.

Reply: The ELISA used in this study targets only the RBD region of spike protein. Therefore, instead of using commercial products, we constructed in-house ELISA. We adopted a value that added twice the standard deviation to the average OD value of plasma from unvaccinated individuals used as negative controls as the cutoff.

Revision (Page 6, Lines 144-146): The cutoff value of OD = 0.3 was determined based on the OD values of plasma from unvaccinated individuals used as a negative control, specifically by adding twice the standard deviation to the average OD value.

How much time passed between the blood sample of the 6 months post-2nd dose and the 3rd dose of vaccine?

Reply: Thank you for this comment. The third vaccination for liver transplant recipients was administered a median of 41 days with an interquartile range (IQR) of 32.5-57.75 days from 6 months after the second vaccination. The shortest interval was 5 days, and the longest was 87 days. For healthy individuals, the median was 42.5 days with an IQR of 42.5-79.5 days. The shortest interval was 4 days, and the longest was 102 days. We illustrated the relationship between the days from 6 months after 2nd vaccination to 3rd vaccination and the plasma anti-RBD antibody titer one month after the third vaccination in a graph. As the interval between vaccine doses increased, there was a slight tendency for the plasma anti-RBD antibody titer to decrease, but the difference was not statistically significant. Therefore, we determined that the difference in interval days did not have a significant impact on our analysis.

What dose of steroids were the patients receiving?

Reply: The patients were taking a median dose of 5 mg/day of steroids (range: 0.5 - 10). Specifically, one patient was taking 0.5 mg/day, six patients were taking 2.5 mg/day, nine patients were taking 5 mg/day, and one patient was taking 10 mg/day. We plotted the steroid dose against the antibody titer and CD4, CD8 T-cell responses one month after the third vaccination on graphs. The antibody titer showed a significant negative correlation with the steroid dose, but no significant correlation was observed for CD4 and CD8 T-cell responses. In a previous study analyzing T-cell responses after mRNA vaccine administration in patients with systemic autoimmune rheumatism, it was shown that those taking a high dose of steroid (>10 mg/day) had significantly weaker CD4 T-cell responses compared to those not taking or those taking a low dose of steroid (< 10 mg/day) (Maliah et al, *Sci. rep.*, 2022). The doses taken by the patients in our study did not exceed 10 mg/day, and the lack of effect on T-cell responses is consistent with the previous study. We have uploaded the graphs as Supplementary Figure 7.

Revision (Supplementary Figure 7 and Page 18, Lines 446-448): Furthermore, the median steroid dose administered to the liver transplant recipients in this study was 5 mg/day (range: 0.5–10 mg/day). Within this range, an effect was observed on the antibody titer, but no effect on the T-cell response was noted (Supplementary Fig. 7A).

How many patients had a humoral response after the 2nd dose and how many of the seronegatives were rescued with the 3rd dose?

Reply: After the second vaccination, anti-RBD antibodies in plasma were induced in 49 out of 54 LTRs. The 5 LTRs in whom anti-RBD antibodies were not induced after the second vaccination all showed induction of the antibodies after the third vaccination. However, there was one individual who, despite having a positive plasma anti-RBD antibody titer after the second dose, did not benefit from the third booster dose and tested negative. This individual was taking three medications: CNI, MMF, and steroids (5 mg/day), and had a low anti-RBD antibody titer even after the second vaccination.

Revision (Page 10, Line 222-228): After the second vaccination, anti-RBD antibodies in plasma were induced in 49 of 54 LTRs. The 5 LTRs in whom anti-RBD antibodies were not induced after the second vaccination all showed induction of the antibodies after the third vaccination. However, there was one individual who, despite having a positive plasma anti-RBD antibody titer after the second dose, did not benefit from the third booster dose and tested negative. This individual was taking three medications, namely CNI, MMF, and steroids (5 mg/day), and had a low anti-RBD antibody titer even after the second vaccination.

No patient during follow-up had a SARS-CoV-2 infection?

Reply: Individuals who were clinically diagnosed with SARS-CoV-2 infection during follow-up were not included in this study.

It is not clear when was considered to have a positive CD4 or CD8 T cell response

Reply: The frequencies of CD154⁺CD4 T cells, Th1, Th2 and CD69⁺4-1BB⁺CD8 T cells were calculated by subtracting the background of unstimulated samples (DMSO). Positive responses were defined if there was a reactivity of 0.01% or more after background subtraction from the unstimulated condition. The following sentences have been added to the Material and Methods section.

Revision (see Page 7, Lines 176-179): Frequencies of CD154⁺CD4 T cells, Th1, Th2 and CD69⁺4-1BB⁺CD8 T cells were calculated by subtracting the background of unstimulated samples (DMSO). Positive responses were defined if there was a reactivity of 0.01% or more after background subtraction from the unstimulated condition.

The discussion in general and the conclusions require a deeper approach and a greater contrast with other published works.

Reply: We have revised the Discussion section to provide a deeper as per your comment.

Revision: Furthermore, passive immunotherapy such as HBIG is mainly used in LTR because HBV vaccine is less effective in LTR²⁷. While the HBV vaccine is a recombinant protein vaccine, the mRNA vaccine developed as a new modality during the SARS-CoV-2 pandemic is a potentially effective platform for inducing neutralizing antibodies even in immunosuppressed LTR. (Page 15, Line 377-381)

CNIs inhibit calcineurin, resulting in the inactivation of the nuclear factor of activated T-lymphocytes (NFAT) and suppression of IL-2 production²⁸. In other words, CNIs specifically target T cells, but for the evaluation of mRNA vaccine effects in LTR, the focus is on humoral immunity due to technical limitations. (Page 15, Line 382-385)

Recent studies suggest that mRNA vaccines induce weaker CD8 T-cell responses in healthy donors compared to CD4 T-cell responses^{38,39}. In solid organ transplant recipients, including liver and kidney transplantation, the frequency of spike-specific CD8 T cells was found to be significantly lower after 2nd mRNA vaccination compared to healthy donors⁹. This finding also indicates that solid organ transplant recipients have weaker CD4 and CD8 T cell responses compared to other groups of immunocompromised patients, such as primary immunodeficiency syndromes, AIDS, hematopoietic stem cell transplant recipients, and chronic lymphocytic leukemia patients. In our study, the frequency of spike-specific CD8 T cells was significantly lower than that of healthy donors at all time points. Although spike-specific CD8 T-cell responses decreased quantitatively in CNI group and CNI+other drug(s) group compared to healthy donors, there is no significant difference between healthy donors and LTR in terms of the booster effect of mRNA vaccines (Figure 3B). (Pages 16-17, Line 411-422)

In recent years, it has been reported that the expression of GZMB and perforin in bulk CD8 T cells, not antigen-specific CD8 T cells, in SARS-CoV-2 infected patients is higher than that in healthy donors³⁸. This suggests that the induction of CD8 T cells expressing high levels of cytotoxic molecules contributes potentially to the suppression of COVID-19 severity. However, a recent study using MHC multimers has demonstrated that the expression levels of GZMB and perforin in SARS-CoV-2 spike epitope-specific CD8 T cells decrease over time after 2nd mRNA vaccination, and do not increase after 3rd vaccination⁴¹. Therefore, the ability of mRNA vaccines to induce CD8 T cells with high cytotoxic activity is limited. (Page 17, Line 423-430)

In summary, no improvement was observed in both the quantitative or qualitative aspects of spike-specific CD8 T cells even after 3rd vaccination in both healthy donors and LTR in our study. (Page 17, Line 432-434)

Graphics are confusing and not self explanatory.

Reply:

Thank you for the comment. We have increased the resolution of the figure as much as possible and made adjustments for better clarity.

Reviewer #4 (Remarks to the Author):

Thank you for inviting me to review this article.

Nogimori and Nagatsuka et al. prospectively evaluated humoral and cellular immune memory response among liver transplant recipients who were booster-vaccinated with mRNA vaccines.

Comments/suggestions

Although there have been few studies in the literature, the authors of the current manuscript have performed exhaustive work on cellular and humoral response assessment and performed several sub-group analysis.

The authors assessed at 1 month after the booster dose. Immune response is expected to be high in the early period. But how long the immunity lasts after the booster is not addressed in the study, which is the actual lacunae in the literature. And how many individuals developed breakthrough infections? this has not been answered.

Reply: After the third vaccination, 11 liver transplant recipients and 2 healthy individuals experienced breakthrough infections.

The time between vaccinations (1,2 and 3rd dose) is not mentioned anywhere in the manuscript. It may be possible that individuals who receive early booster (shorter gap between 2nd and 3rd dose) after 2nd dose have a high response. Or did everybody in LTR and HD receive all vaccinations at the same time?

Reply: The third vaccination for liver transplant recipients was administered a median of 41 days with an interquartile range (IQR) of 32.5-57.75 days from 6 months after the second vaccination. The shortest interval was 5 days, and the longest was 87 days. For healthy individuals, the median was 42.5 days with an IQR of 42.5-79.5 days. The shortest interval was 4 days, and the longest was 102 days. We illustrated the relationship between the days from 6 months after 2nd vaccination to 3rd vaccination and the plasma anti-RBD antibody titer one month after the third vaccination in a graph. As the interval between vaccine doses increased, there was a slight tendency for the plasma anti-RBD antibody titer to decrease, but the difference was not statistically significant. Therefore, we determined that the difference in interval days did not have a significant impact on our analysis.

I assume everyone was a homologous recipient of vaccination. Future studies could evaluate the impact of heterologous vaccination, which has been reported to be more beneficial.

Reply: Unfortunately, in Japan, heterologous vaccine regimens have not yet been approved, and all participants in this study received homologous vaccination.

The authors report lack of CD8 response after booster.

Figure 1E/H: The authors mention that anti-RBD and pVNT50 in LTR reach comparable levels with HD after 3rd dose. (1 month after 2nd dose in the healthy group vs. 1 month after 3rd dose in LTR group.) The endpoints are different. What was the comparison at 1 month post 3rd dose among these groups? I may have missed it probably.

Reply: We apologize for the unclear notation. The comparison one month after the third dose is presented in Figure 1D/G.

Line 259/260 authors claimed that “Furthermore, Th1/Th2 ratio in LTRs was significantly lower compared to HDs (Figure 2G), suggesting that LTRs are more susceptible to the induction of Th2-biased CD4+T-cell responses”. In general, LTRs with immune suppressive drugs usually have Th2 lineage and more Th2 cytokines compared to Th1.

Reply: Thank you for the comment about an important point. I added the following sentences to the Discussion section.

Revision (Page 16, Lines 405-406): Furthermore, previous studies have demonstrated a clear shift from Th1

to a Th2 cytokine-secreting profile as an additional mechanism of immunosuppression by cyclosporin A

and steroids.

Throughout the study authors have compared all the parameters between healthy and LTRs (CNI/CNI+other drugs) at different time points. This is important although it would have been interesting to learn the comparison between two groups CNI alone (n=23) vs. CNI +other drugs (n=31). Data can be analysed within the same group at different time (5 time points) points with statistical analysis. Also, mention this n of each group somewhere in the results. i.e., 23 vs. 31 vs. 44. Similarly, I don't see a number (sample size) anywhere for all sub-group analysis. For example: comparing LDLT vs. DDLT; recipients 12 years after and less. Why this 12 years was chosen?

Reply: We apologize for the lack of sample sizes. We have included the sample size (N) in the Figure legend according to the reviewer's suggestion.

Additionally, regarding the categorization based on the number of years post-transplantation, we adopted the median number of years since transplantation for this cohort of transplant recipients.

MMF and mTORi have been reported to affect the immune response and it is suggested to stop this drug in infected individuals (PMID: 32750442; PMID: 36374707). Although this hypothesis is contentious it would be interesting to know if there was any differential expression of immune response in those on MMF/mTORi.

Reply: A graph comparing antibody titers of LTRs further divided into two groups according to whether they were taking MMF or not was created and added as a new Supplementary Figure 1D, and the following text was added to the results.

As for mTORi, only three people were taking it.

Revision (Supplementary Figure 1D and Page 9, Lines 234-235): LTRs who have taken MMF also exhibited lower antibody titers than those who have not (Supplementary Fig. 1D).

Did you analyse Th1/Th2 ratio between LTR and HD pre and post which would have added more value to the current study.

Reply: Figure 2G shows the Th1/Th2 ratio at 1-month after the second and third vaccinations, and a new 6-month time point after the second vaccination has been added as per your comment (Figure 2G).

CD 4 and 8 naïve cells in the marrow of LTR are unaffected, while the CD4 effector memory and CD4 and 8 effector cells are lower in LTRs which has been reported in the literature. Similarly, the authors report that after

booster dose CM changes to EM. However, on Page no 24, Figure 3, data for CD8 memory cells of 3rd post 1-month vaccination is missing both. Did the authors perform the analysis or missed?

Reply: We apologize for missing data for CD8 memory cells of 3rd post 1-month vaccination. We added the data as new Figures 3D and 3E and the following sentence in the result section. (See Page 12, Line 305-307)

After 3rd mRNA vaccination, HDs and LTRs showed different phenotypes of spike-specific CD8⁺ T cells, with decreased CM and increased EM and Effector in HDs, but a trend toward increased CM in LTRs.

There are several panels (A-H). Please number them sequentially in the manuscript in bold letters.

Reply: We have formatted the figure numbers in bold as per your comment and have rearranged the figure numbering sequentially in the order of mention in the revised manuscript.

Line 289: over- r is missing

Reply: We have fixed this in the revised manuscript.

REVIEWERS' COMMENTS:

Reviewer #1 (Remarks to the Author):

Dear Nogimori and Nagatsuka,

Thank for yours answers, explanations and exhaustive work. I agree with the improvements.

Reviewer #4 (Remarks to the Author):

Changes noted. Thank you